# Unveiling the Potential Ways to Apply Citrus Oil to Control Causative Agents of Pullorum Disease and Fowl Typhoid in Floor Materials

**DOI:** 10.3390/ani14010023

**Published:** 2023-12-20

**Authors:** Dita Julianingsih, Chuan-Wei Tung, Kanchan Thapa, Debabrata Biswas

**Affiliations:** 1Department of Animal and Avian Sciences, University of Maryland, College Park, MD 20742, USA; djuliani@umd.edu (D.J.); vmtung@umd.edu (C.-W.T.); kthapa@umd.edu (K.T.); 2Biological Sciences Program, University of Maryland, College Park, MD 20742, USA

**Keywords:** microbiology, natural antimicrobial, citrus oil, salmonellosis, infectious diseases

## Abstract

**Simple Summary:**

In this study, the utilization of citrus oil (CO) as a natural antimicrobial solution to address bacterial diseases in poultry farming was examined, with a focus on pullorum disease and fowl typhoid in wooden chips employed as floor materials. With a decline in antibiotic use in poultry leading to a resurgence of bacterial infections like salmonellosis, CO has emerged as a promising alternative. The oil effectively inhibited the growth of various Salmonella serovars, particularly *S. Gallinarum* and *S. Pullorum*, demonstrating its potential to control these diseases. This study also found that CO, when applied to wooden chips in poultry house floors, prevents the development of Salmonella. Additionally, the research revealed that CO downregulates virulence genes in the bacteria, suggesting it could alter their harmful effects. Overall, this natural antimicrobial shows promise in preventing and managing salmonellosis in chicken production, offering a valuable alternative for addressing re-emerging diseases and promoting healthier poultry farming practices.

**Abstract:**

This study investigates the potential role of Cold-pressed Valencia Terpeneless citrus oil (CO), as a natural antimicrobial, in controlling causative agents of pullorum disease and fowl typhoid in floor materials for poultry farming, specifically wooden chips. The study addresses the issues that have arisen as a result of the reduction in antibiotic use in poultry farming, which has resulted in the re-emergence of bacterial diseases including salmonellosis. CO efficiently inhibits the growth of pathogens including various serovars of *Salmonella enterica* (SE), including SE serovar Gallinarum (*S. Gallinarum*) and SE serovar Pullorum (*S. Pullorum*), in a dose-dependent manner. Minimum Inhibitory Concentration (MIC) and Minimum Bactericidal Concentration (MBC) of CO showed potential for controlling diverse *S. Gallinarum* and *S. Pullorum* isolates. Growth inhibition assays demonstrated that 0.4% (*v*/*w*) CO eliminated *S. Pullorum* and *S. Gallinarum* from 24 h onwards, also impacting poultry gut microbiota and probiotic strains. Floor material simulation, specifically wooden chips treated with 0.4% CO, confirmed CO’s effectiveness in preventing *S. Gallinarum* and *S. Pullorum* growth on poultry house floors. This study also investigated the effect of CO on the expression of virulence genes in *S. Gallinarum* and *S. Pullorum*. Specifically, the study revealed that the application of CO resulted in a downregulation trend in virulence genes, including *spiA*, *invA*, *spaN*, *sitC*, and *sifA*, in both *S. Pullorum* and *S. Gallinarum*, implying that CO may alter the pathogenicity of these bacterial pathogens. Overall, this study reveals that CO has the potential to be used as a natural antimicrobial in the prevention and management of *Salmonella*-related infections in chicken production, offering a viable alternative to control these re-emerging diseases.

## 1. Introduction

Poultry farming has undergone significant transformations over the years to meet the demands of a growing global population for safe, healthy, and affordable animal protein [1]. This evolution includes adapting to changing consumer preferences and addressing public health concerns [2,3]. One notable transformation in poultry farming practices has involved either reducing antibiotic usage, particularly in subtherapeutic doses within conventional poultry farming, or completely eliminating antibiotics, as seen in organic poultry farming. This shift is a response to the escalating threat of antibiotic resistance [4]. Antibiotics were historically used extensively in poultry farming to promote growth and prevent bacterial diseases [5]. However, the indiscriminate use of antibiotics has led to the emergence and spread of antibiotic-resistant bacteria, posing risks to both human and animal health [6,7].

As a result, many countries have implemented regulations to reduce or eliminate antibiotic use in livestock production [8]. While this is a positive step in combating antibiotic resistance, it has presented challenges within the poultry industry [9]. The decrease in antibiotic usage has contributed to the re-emergence of bacterial diseases, such as fowl typhoid and pullorum disease, which were previously under control [10,11]. These diseases, caused by *Salmonella enterica* serovar Gallinarum and serovar Pullorum, respectively, lead to significant economic losses due to high mortality rates [12,13]. Considering this re-emerging situation, the exploration of natural antimicrobials could be a potential solution to prevent and control these diseases [14].

Pullorum disease primarily affects young birds, while fowl typhoid extends its impact to growing and adult poultry. The severity varies, influenced by factors like the bird’s breed and age, with mortality rates reaching up to 100% in highly susceptible birds. Although eradicated from commercial poultry in developed countries like the USA, Canada, Australia, and Europe, these diseases persist in backyard flocks and game birds. Control is challenging, especially in game birds raised in semi-wild systems. Occasional outbreaks occur in industrial poultry, raising concerns about increased exposure with the growing demand for free-range birds [15,16,17]. In response to the challenges posed by antibiotic resistance and the re-emergence of bacterial diseases, the exploration of natural antimicrobials has gained significant interest [18]. Natural antimicrobials are derived from various sources such as plants, animals, and microorganisms, and they possess inherent antimicrobial properties [19]. One such natural antimicrobial that has shown promise in preventing *Salmonella* is citrus oil (CO), derived from citrus fruits such as oranges and lemons [20].

Numerous studies have been conducted to investigate the potential of CO as a natural antimicrobial in the prevention and control of bacterial infections. CO contains bioactive chemicals, such as limonene and linalool, which have been shown to have antimicrobial properties against a wide range of pathogens. These include notable pathogens such as *Streptococcus aureus*, *Listeria monocytogenes*, *Campylobacter jejuni*, *Arcobacter* spp., and *Salmonella* spp. [20,21,22,23,24,25,26,27]. These compounds can disrupt bacterial cell membranes, interfere with enzymatic activity, and disrupt bacterial metabolism, leading to bacterial death or growth inhibition [27,28,29]. This study investigated the possibility of CO as a natural antimicrobial for preventing pullorum disease and fowl typhoid. In simulated poultry environments, the interactions of CO with *S. Pullorum* and *S. Gallinarum* were examined. Additionally, the aim was to assess the impact of CO at various concentrations on the gut microbiome of chickens and probiotic bacteria.

## 2. Materials and Methods

### 2.1. Bacterial Strains and Culture Conditions

In this study, we used two *Salmonella enterica* (SE) serovars, which are pathogenic to poultry, specifically SE serovar Gallinarum (CAT 375, Presque Isle Cultures, Erie, PA, USA) and SE serovar Pullorum (ATCC 13036). Additionally, we used *S. Pullorum* and *S. Gallinarum* samples which were isolated previously from farm environments or poultry products in our lab. Three common chicken gut microbiomes, including *Enterococcus faecalis* (PIC 522A), *Streptococcus thermophilus* (ATCC 19258), and *Lactobacillus helveticus* (ATCC 8018), were used. As a probiotic reference, *Escherichia coli* Nissle 1917 (Mutaflor, Herdecke, Germany) was employed. Before each experiment, *S. Gallinarum*, *S. Pullorum*, and *E. coli* Nissle were cultivated and maintained on Luria-Bertani (LB) agar (Millipore, Billerica, MA, USA); *L. helveticus* was cultured in de Man Rogosa Sharpe (MRS) agar (Merck, Germany); and *E. faecalis* and *S. thermophilus* were cultured in Tryptic Soy (TS) agar (Hardy Diagnostic, Santa Maria, CA, USA) plates, all at 37 °C for 18–24 h from −80 °C glycerol stock. The bacterial strains were stored at −80 °C in glycerol, and prior to each experiment, they were cultured on their respective agar plates for the specified duration.

### 2.2. CO and Working Solution Preparation

This study utilized Cold-pressed Valencia Terpeneless citrus oil (Firmenich, Safety Harbor, FL, USA) as a natural antimicrobial. A 20% ethanol solution was employed as a diluent for the CO, ensuring that any observed effects were not due to the ethanol component [30]. A separate control experiment utilizing only the 20% ethanol solution revealed that there was no reduction of *S. Pullorum* and *S. Gallinarum* growth. This demonstrates that the growth inhibition found in this experiment is purely attributable to the presence of CO, identifying it as the active antimicrobial agent in the study.

### 2.3. Determination of Minimum Inhibitory and Bactericidal Concentration of CO against S. Gallinarum and S. Pullorum

The broth dilution technique described by Federman et al. with modification was used to determine the MIC [25]. CO with concentrations ranging from 0.025% to 0.8% (*v*/*v*) was evaluated for its ability to inhibit the growth of *S. Gallinarum* and *S. Pullorum*. The OD of *S. Pullorum* and *S. Gallinarum* cell suspensions was standardized to 0.1 at OD_600nm_, containing approximately 2 × 10^6^ CFU/mL. These concentrations of diluted CO were added to a 24-well plate (Greiner Bio-One, Monroe, NC, USA) containing either 100 µL of *S. Gallinarum* or *S. Pullorum*. To determine the Minimum Inhibitory Concentration (MIC), plates were incubated at 37 °C for 24 h to assess the inhibition of growth in *S. Gallinarum* and *S. Pullorum*. To calculate the MBC as described previously by Birhanu et al. [31], 10 µL of the suspension from the microplate was collected and plated to LB agar starting from the MIC value. The plates were incubated at 37 °C for 24 h to identify any potentially slow-growing microorganisms. All experiments were repeated 4 times.

### 2.4. Growth Inhibition Assay of Poultry Bacterial Pathogens, Gut Microbiome, and Probiotic Strains

To conduct the growth inhibition assay in broth, *S. Gallinarum*, *S. Pullorum*, and *E. coli* Nissle were cultured in LB broth while *E. faecalis* and *S. thermophilus* were cultured in TS broth, and *L. helveticus* was cultured in Lactobacilli MRS broth (Hardy Diagnostic, Santa Maria, CA, USA), all maintained at 37 °C for 24 h. The optical density (OD) of the bacterial cell suspensions was standardized to 0.1 at OD_600nm_ using a Multiskan FC spectrophotometer (Thermo Fisher Scientific, Shanghai, China). In 50 mL tubes, LB, TS, or MRS broth with varying concentrations of CO (0.1%, 0.2%, 0.3%, and 0.4%, *v*/*v*) were prepared by adding the necessary volume of extracts from stock solutions. Control wells received autoclaved deionized water instead of the extracts. Bacterial suspension (200 µL), containing approximately 2 × 10^6^ CFU/mL, were introduced into each tube. The tubes were then incubated for different time intervals (24, 48, and 72 h) at 37 °C with agitation at 120 rpm. Following incubation, bacterial suspensions underwent serial dilution in PBS (Phosphate buffer saline) and were plated on specific agar media: LB agar for *S. Gallinarum*, *S. Pullorum*, and *E. coli* Nissle; MRS agar (de MAN, ROGOSA and SHARPE) (Merck, Darmstadt, Germany) for *L. helveticus*; and TS agar for *E. faecalis* and *S. thermophilus*. This serial dilution process was performed to quantify the colony-forming units per milliliter (CFU/mL).

### 2.5. Determine the Growth of S. Pullorum and S. Gallinarum in Simulated Floor Material, Wooden Chips Treated with CO

Containers filled with 5 g of wooden chips (NEPCO, Warrensburg, NY, USA) were prepared. The OD of *S. Pullorum* and *S. Gallinarum* cell suspensions was standardized to 0.1 at OD_600nm_. Approximately 2 × 10^5^ CFU/mL of *S. Pullorum* and *S. Gallinarum* in PBS was sprayed (separately) onto the wooden chips with a total volume of around 10 mL. The wooden chips and the inoculates were mixed thoroughly. 0.4% CO was then sprayed onto the container, followed by thorough mixing. Autoclaved deionized water was added to the control container, replacing the CO. The containers were incubated for different time intervals (24, 48, and 72 h) at 37 °C. Subsequently, serial dilutions in PBS were performed, and the samples were plated on LB agar to determine the CFU/mL. The experiment was repeated 4 times to ensure the results were consistent.

### 2.6. The Effect of CO on the Gene Expression of S. Gallinarum and S. Pullorum Virulence Genes in Culture Condition and on Wooden Chip

Methods were adapted from those previously described in the literature by Alvarado-Martinez et al. [32]. Inoculate obtained from growth inhibition assay samples at 24 h, featuring diverse CO concentrations (0.1%, 0.2%, and 0.3%), was employed. In the case of broth samples, the inoculate underwent centrifugation at 4000× *g* for 10 min to collect the pellet. For the wooden chip samples, DNA isolation commenced using the DNeasy PowerSoil Pro Kit (QIAGEN, Hilden, Germany). Subsequently, both broth and wooden chip samples underwent a meticulous process of RNA extraction, conducted with precision and utilizing the Trizol reagent (Molecular Research Center Inc., Cincinnati, OH, USA). RNA concentration was quantified with a NanoDrop spectrophotometer (Thermo Fisher Scientific Inc., Marietta, OH, USA) for standardization. The standardized RNA served as the template for cDNA synthesis, achieved with the High-Capacity cDNA Reverse Transcription Kit (Applied Biosystems, Foster City, CA, USA), following the instructions provided by the manufacturer. The cDNA synthesis protocol included incubation at 25 °C for 10 min, 37 °C for 120 min, and 85 °C for 5 min. Quantification of changes in the relative expression of genes associated with the virulence of *S. Pullorum* and *S. Gallinarum* was conducted using a qPCR machine (Illumina, Singapore). The qPCR reaction was prepared according to the PerfeCTa SYBR Green Fast Mix protocol (Quanta Bio, Beverly, MA, USA), and executed using the Eco Real-Time PCR system (Illumina, San Diego, CA, USA). The cycle protocol consisted of 30 s at 95 °C, followed by 40 cycles of 5 s at 95 °C, 15 s at 55 °C, and 10 s at 72 °C. Custom primer sequences (Table 1), aligning with the conserved regions of respective virulence genes in *S. Pullorum* and *S. Gallinarum*, were utilized for comparison against a reference house-keeping gene belonging to 16S-rRNA. 

### 2.7. Statistical Analysis 

PrismPad version 8.2.0 from GraphPad Software, Inc., La Jolla, CA, USA, was employed to assess the statistical significance between treatments for *S. Gallinarum* and *S. Pullorum*. The analysis included a Student’s *t*-test, and multiple *t*-tests were also utilized to confirm the significant difference (*p* < 0.05) between the untreated control and the distinct treatment with each compound [32].

## 3. Results

### 3.1. MICs and MBCs of CO to Inhibit S. Pullorum and S. Gallinarum

The MIC and MBC of CO against *S. Gallinarum* and *S. Pullorum* were identified as identical for each of the evaluated isolates in Table 2. The MIC and MBC of CO for only *S. Gallinarum* CAT375 were both 0.2%, showing that this concentration of CO is efficient in both inhibiting and eliminating the bacterium. However, the MIC and MBC values of CO for both *S. Pullorum* strains (ATCC and farm isolates), as well as *S. Gallinarum* organic retail chicken isolate, were 0.4%, indicating that this concentration (0.4%) of CO is required to inhibit the growth of both of these bacterial pathogens (Table 2). Consistent MIC and MBC values affirm the antimicrobial properties of the tested substance. Collectively, the data indicate that the tested CO is effective in eliminating these *Salmonella* bacterial isolates at the specified concentrations.

### 3.2. Inhibitory Effects of CO on Poultry Bacterial Pathogens, Gut Microbiome, and Probiotic Strains in Broth Media

Growth inhibition assays of *S. Pullorum* and *S. Gallinarum* were performed over 72 h at varying concentrations of CO (Figure 1). The inhibitory effects of CO were observed for both serovars of *Salmonella,* as increasing concentration of CO in the treatment led to reduced growth in CFU/mL relative to the control. CO at a concentration of 0.4% completely inhibited the growth of both *S. Pullorum* and *S. Gallinarum* from 24 h of treatment and beyond, as growth was not within detectable limits. For both serovars, there was a significant difference in inhibition of growth between 0.2% and 0.3% CO treatment as compared to control (*p* < 0.05). The difference between the two concentrations was large, as the 0.3% treatment had around 5 logs less in CFU/mL than the 0.2% treatment. Similarly, there was not a significant difference in terms of growth between the control and the 0.1% CO treatment groups as the difference in CFU/mL was around 1–2 logs. These findings collectively suggest that 0.4% CO at a treatment time of 24 h or greater may be more effective in inhibiting growth of *S. Gallinarum* and *S. Pullorum*.

This study also presents detailed data on the time-dependent inhibitory effects of CO on various poultry gut microbiota, including *L. helveticus*, *E. faecalis*, *S. thermophilus*, and the probiotic *E. coli* Nissle, within a broth medium environment. At the initial time point (0 h), all concentrations of CO, ranging from 0.1% to 0.4%, exhibit comparable CFU/mL when compared to the control group, suggesting no immediate impact. However, as time progresses, the inhibitory effects become increasingly evident. For *L. helveticus*, the log CFU/mL decreases at 24 h for all concentrations, with 0.4% CO showing the most pronounced effect. The inhibitory effects become most noticeable by 48 and 72 h, especially at greater concentrations, which significantly reduces the growth of bacteria. Similar trends are observed for *E. faecalis*, where the inhibitory effects become more apparent over time. At 24 h, all concentrations result in a decrease in log CFU/mL compared to the control, with higher concentrations exhibiting stronger inhibitory effects. This pattern persists at 48 and 72 h, with 0.4% concentration demonstrating a substantial impact on reducing *E. faecalis* population. In the case of *S. thermophilus*, inhibitory effects are noticeable as early as 24 h, with higher concentrations leading to a more significant reduction in log CFU/mL. Inhibitory effects develop after 48 h, and at 72 h, 0.4% concentration totally inhibits *S. thermophilus* growth. For *E. coli* Nissle, inhibitory effects are apparent at 24 h, with higher concentrations exhibiting a more substantial impact. Inhibitory effects become more noticeable after 48 and 72 h, particularly at 0.4% concentration, which fully inhibits bacterial growth.

### 3.3. Time-Dependent Inhibitory Effects of CO on S. Pullorum and S. Gallinarum Growth in Environmental Simulations

The decision to incorporate a 0.4% concentration of CO in the environmental simulation was driven by the findings presented in Figure 1, where this concentration demonstrated complete inhibition of both *S. Pullorum* and *S. Gallinarum* in broth media. Under conditions simulating the poultry environment, a significant inhibitory effect on the growth of *S. Gallinarum* and *S. Pullorum* was noted in the presence of CO (Figure 2). When using a 0.4% concentration of CO, a remarkable reduction in *S. Gallinarum* growth in wooden chip by around 5, 4, and 4 logs at 24, 48, and 72 h, respectively (Figure 2A), was observed. Similarly, this same concentration of CO caused a substantial decrease in *S. Pullorum* growth, by 4, 5, and 5 logs at 24, 48, and 72 h, respectively (Figure 2B). The bacterial reduction is comparatively more pronounced at the 24-h time point in the case of *S. Gallinarum*. Conversely, an increased bacterial reduction is evident after 48 and 72 h when compared to the reduction observed after 24 h in *S. Pullorum*.

### 3.4. Effect of CO on Expression of S. Gallinarum and S. Pullorum Virulence Genes

Across all concentrations evaluated (0.1%, 0.2%, 0.3%) in broth media, a consistent reduction in the expression levels of *spiA*, *invA*, *sitC*, *sipB*, and *sifA* in *S. Gallinarum* was observed. Notably, the expression of *spiA* and *sipB* in *S. Gallinarum* showed a significant decrease at 0.3% CO treatment (*p* < 0.05). In contrast, the impact of CO treatment on *spaN* expression varied. At 0.1%, *spaN* expression showed minimal impact, but increased concentrations of CO (0.2% and 0.3%) resulted in a subsequent decrease in *spaN* expression levels (Figure 3A). 

In *S. Pullorum* trials, a downregulation trend across all tested concentrations of CO in *spiA*, *invA*, *spaN*, *sitC*, and *sifA* was observed. Remarkably, *invA* and *sitC* were significantly downregulated at 0.1% CO treatments (*p* < 0.05). *SipB*, however, maintained an upregulated expression profile throughout CO treatments (Figure 3B).

The expression of virulence genes in *S. Gallinarum* (Figure 4A) and *S. Pullorum* (Figure 4B) was investigated in this study under different conditions. The control group, represented by a baseline of 0, was subjected to a simulated environment with water, whereas the treatment group was subjected to identical conditions with the addition of 0.4% CO. The relative expression levels of specific genes were measured in log-fold changes. When *S. Gallinarum* (Figure 4A) was treated with 0.4% CO, the genes *spiA*, *invA*, *spaN*, *sitC*, and *sipB* showed a significant decrease in expression (*p* < 0.05), with values ranging from −2.23 to −4.16. This implies that the presence of CO in the environment inhibits the expression of these virulence genes in *S. Gallinarum*. When *S. Pullorum* (Figure 4B) was treated with 0.4% CO, the genes *spiA*, *invA*, *spaN*, and *sitC* showed reduced expression levels, with *spiA* and *spaN* exhibiting statistically significant decreases in expression (*p* < 0.05). However, there were no substantial changes in the expression of *sipB* or *sifA*. This suggests that CO has a limited effect on *S. Pullorum* virulence gene expression. This study reveals that applying 0.4% CO to the environment alters the expression of virulence genes in both *Salmonella* strains, with *S. Gallinarum* showing more pronounced effects than *S. Pullorum*.

## 4. Discussion

According to the literature, the quantitative and qualitative analysis of Cold-pressed Terpeneless Valencia citrus oil, as utilized in this study, reveals a significant composition primarily dominated by linalool, comprising 20.2% of the fraction. Following closely are decanal at 18%, citral at 14.1%, geranial at 9.1%, α-terpineol at 5.8%, valencene at 5.2%, neral at 5.0%, dodecanal at 4.1%, citronellal at 3.9%, and limonene at 0.3%. Notably, linalool emerges as the dominant constituent upon gas chromatography and mass spectrometry (GC–MS) analysis. Moreover, decanal and geranial have been previously recognized for their antimicrobial properties. This comprehensive breakdown sheds light on the distinctive chemical profile of Cold-pressed Terpeneless Valencia citrus oil and highlights potential functional attributes associated with its major constituents [23,25]. Further, citral and linalool have shown their strong antimicrobial properties, as have many other terpenes; their mechanism of action is thought to be due to bacterial cell membrane injury, which results in K^+^ ion leakage and decreased membrane potential [35]. 

Guo et al. [29] investigated the antibacterial activity and potential mechanism of action of linalool against *Pseudomonas fluorescens*. The observed reduction in membrane potential (MP), leakage of alkaline phosphatase (AKP), and the release of macromolecules, including DNA, RNA, and proteins, provided evidence that the treatment with linalool resulted in damage to the cell wall membrane structure and leakage of cytoplasmic contents. Additionally, the decrease in enzyme activity, specifically succinate dehydrogenase (SDH), malate dehydrogenase (MDH), pyruvate kinase (PK), and ATPase, suggested that linalool induced metabolic dysfunction and inhibited energy synthesis. Moreover, the impact on the activity of respiratory chain dehydrogenase and the metabolic activity of respiration pointed to the ability of linalool to inhibit cellular respiration. Collectively, these findings indicated that linalool exhibits potent antibacterial activity against *P. fluorescens* by causing membrane damage, disrupting bacterial metabolism, inducing oxidative respiratory perturbations, interfering with cellular functions, and ultimately leading to cell death. 

Han et al. [28] conducted a study unveiling the antimicrobial activity and mechanism of limonene against *Staphylococcus aureus*. The study employed various methodologies, including scanning electron microscopy (SEM), assessment of alkaline phosphatase (AKP) activity reduction, and fluorescence microscope observations, all confirming the ability of limonene to disrupt the cell morphology and cell wall integrity of *S. aureus*. Fluorescein diacetate staining experiments revealed a reduction in mean fluorescence intensity (MFI), signifying damage to the cell membrane and increased membrane permeability induced by limonene. The concurrent decrease in membrane potential (MP) further substantiated the membrane damage and reduction in respiratory metabolic activity. Respiratory depression tests pinpointed that limonene primarily affected the tricarboxylic acid (TCA) pathway, followed by the Embden-Meyerhof-Parnas (EMP) pathway, influencing the respiratory metabolism of *S. aureus*. Additionally, disturbances in key enzymes such as succinate dehydrogenase (SDH), malate dehydrogenase (MDH), pyruvate kinase (PK), and ATPase (T-ATPase, Na+-K+-ATPase, Ca2+-Mg2+-ATPase), along with alterations in ATP concentration, illustrated the impact of limonene on metabolism and ATP synthesis inhibition.

In a study conducted by O’Bryan et al. [20], the Minimum Inhibitory Concentration (MIC) of CO against various SE serovars, specifically human pathogens including Typhimurium, Enteritidis, Senftenberg, Tennessee, Kentucky, Heidelberg, Montevideo, Michigan, and Stanley, was determined, and they found the MIC values of CO range from 0.125% to 0.5% for different *Salmonella* strains. Consistently with previous research, it was also revealed that CO has antimicrobial activity against both poultry pathogens. This highlights the ability of CO to act as a natural antimicrobial to control salmonellosis in poultry.

In view of these promising results demonstrating the inhibitory effects of CO on the causative agents of pullorum disease and fowl typhoid, it is critical to recognize the potential limitations and considerations for practical application. While the 0.4% concentration of CO displayed significant antimicrobial properties against *S. Pullorum* and *S. Gallinarum* in the simulated poultry environment, it is crucial to highlight that due to its impact on beneficial bacteria, this concentration may not be acceptable for oral administration due to its negative impact on common gut microflora. To verify its effect on normal poultry gut microbiota, the growth of several common poultry gut microbes in the presence of effective concentrations of CO was assessed. The study determined that the same concentration of CO also affected *L. helveticus*, *E. faecalis*, *S. thermophilus*, which are part of the normal gut flora in chickens [36,37,38], and *E. coli* Nissle, a beneficial bacterium for the chicken gut [39]. The broad-spectrum antimicrobial activity observed against both pathogenic and beneficial bacteria suggests a need for caution in direct oral application. The potential impact on beneficial gut flora raises concerns about disrupting the delicate microbial balance essential for poultry health and digestion. As a prudent alternative, spraying CO in the environment has emerged as a realistic option. This approach enables tailored antimicrobial action against environmental pathogens while limiting direct contact with beneficial gut flora. The antimicrobial characteristics of CO could be utilized to establish a more favorable environment for poultry. This approach aims to enhance the poultry house environment without jeopardizing the essential microbial populations within the digestive system. 

This study also investigated the impact of CO on the expression of virulence genes in *S. Pullorum* and *S. Gallinarum*. Notably, when exposed to CO, key virulence genes such as *spiA*, *invA*, *spaN*, *sitC*, *sipB*, and *sifA* exhibited a downregulation trend. Ochman et al. [40] identified a specific area on the *Salmonella* Typhimurium chromosome that is home to important genes like *spiA* and *spiB*. *Salmonella* virulency in mice is dependent on a genetic nexus formed by these genes, which encode proteins similar to those seen in type III secretion systems. The ability of *Salmonella* to survive and replicate within host cells is mostly dependent on the type 3 secretion system 2 (T3SS-2), which is constructed by this genetic ensemble, which includes SPI-2 genes such as *spiA* [41,42]. Moreover, the main factor influencing virulence is the *Salmonella* pathogenicity island 1 (SPI-1) type III protein secretion system (T3SS). According to Lou et al. [43], this system plays a crucial role in *Salmonella* pathogenicity by supplying effector proteins necessary for intestinal invasion and the development of enteritis.

*Salmonella* employs genes within SPI-1, including *invA*, *orgA*, *prgH*, *sipB*, and *spaN*, to encode the type 3 secretion system 1 (T3SS-1), crucial for invading both phagocytic and non-phagocytic cells [41,42]. As part of the *inv* locus identified by Galán et al. [44], *invA*, the first gene in an operon with at least two additional invasion genes, is pivotal in facilitating entry of *Salmonella* into cultured epithelial cells. Shanmugasamy et al. [45] highlighted that the *invA* gene codes for a protein in the inner membrane of bacteria which is essential for invading epithelial cells. According to a study by Mohammed in 2022, the *invA* gene was proposed to play a crucial role in regulating *Salmonella* colonization and invasion processes within the gut of free-range chickens [46].

Several studies have demonstrated the critical role which the *spaN* gene plays in *Salmonella* virulency progression. Zhang et al. [33] emphasize the critical role that *spaN* plays in allowing *Salmonella* to enter nonphagocytic cells, highlighting the significance of *spaN* during the early stages of infection. Furthermore, the role of *spaN* in the type III secretion system is highlighted by Skyberg et al. [34], who identify it as a crucial component of the SPI-1 TTSS needle tip complex. *Salmonella* uses this molecular syringe as a covert tool to inject bacterial proteins straight into host cells. Expanding upon these discoveries, Webber et al. relate the *spaN* gene to *Salmonella* invasiveness, highlighting the ability of the pathogen to infiltrate non-phagocytic cells and elude immune responses, such as macrophage destruction [47]. 

*Salmonella* requires the *sipB* gene to create functional pores during erythrocyte infection in order to enter the host cell through the plasma membrane [48]. The *sipB* gene is known as a translocator because it transports *Salmonella* effector proteins into host cells (RTE shrimp eaters), causing typhoid fever and gastroenteritis [49]. In *Salmonella* serovars, the *sipB* gene promotes apoptotic macrophages either by activating or inducing autophagy and mitochondrial disruption, or by binding the proapoptotic enzyme caspase-1, resulting in the release of interleukin-1 beta active form [50,51]. 

Bacterial membrane proteins linked to iron absorption are encoded by the *sitC* gene. On the other hand, the proteins encoded by the *iroN* gene serve as siderophore receptors. Following cell invasion, *Salmonella* species discover a low-iron environment, which is crucial for their survival and proliferation inside the host cell. As a result, bacteria have created different iron procurement systems; these systems are typically redundant and do not express simultaneously [47].

*SifA*, a critical *Salmonella* effector protein, is transported into infected cells via the pathogenicity island 2-encoded type 3 secretion pathway. It interacts with the eukaryotic protein SKIP (*sifA* and kinesin-interacting protein) to regulate kinesin-1 levels in the *Salmonella*-containing vacuole (SCV). This connection is critical for *Salmonella* virulence because it affects vacuolar membrane dynamics. *SifA* has two domains: the N-terminal, which interacts with the host protein SKIP, and the C-terminal, which is similar to other bacterial effector domains with guanine nucleotide exchange factor activity. The C-terminal domain of *sifA* contributes independently of SKIP, and both domains work together in the signaling cascade that supports *Salmonella* virulence [52,53,54].

## 5. Conclusions

CO is shown in this study to be effective against *S. Pullorum* and *S. Gallinarum*, with consistent MIC and MBC values of 0.2% and 0.4%. At 0.4% concentration, CO effectively inhibits the growth of both bacterial pathogens after 24 h, as demonstrated in this study. Simulated floor environments that replicate the chicken-farming environment exhibit a notable decrease in bacterial growth at a concentration of 0.4% CO. Furthermore, this study explores the expression of virulence genes, observing a trend toward downregulation at different concentrations of CO, thereby increasing its antimicrobial efficacy. For *S. Gallinarum*, 0.4% CO dramatically reduces the expression of *spiA* and *sipB*, whereas *invA* and *sitC* are markedly downregulated in *S. Pullorum*. Therefore, CO has potential for controlling *Salmonella* in chickens as a dual-action drug that inhibits growth of these poultry pathogens and modifies their virulence genes.

## Figures and Tables

**Figure 1 animals-14-00023-f001:**
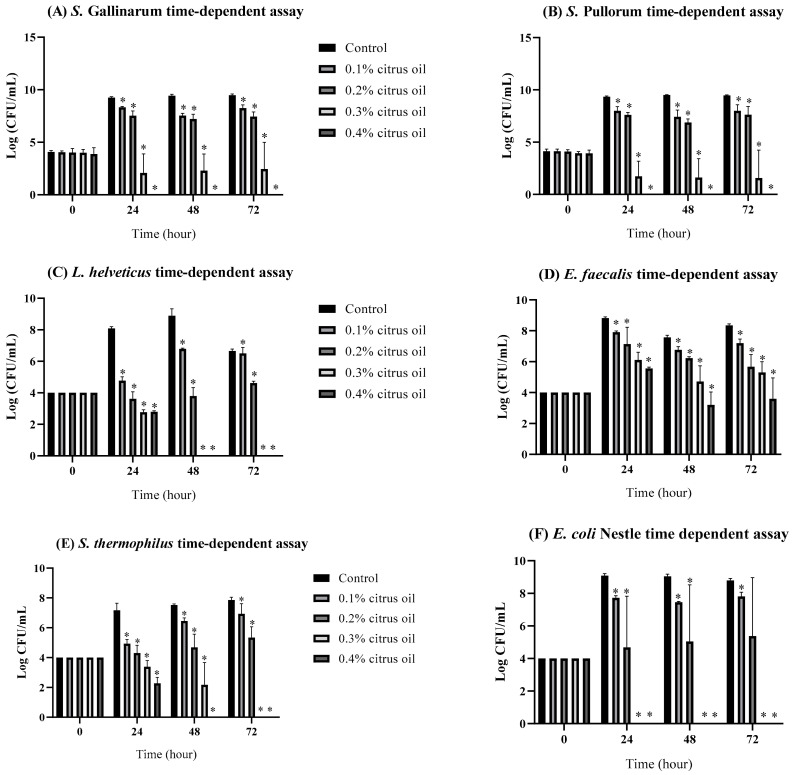
Growth performance of (**A**) *S. Gallinarum*, (**B**) *S. Pullorum*, (**C**) *L. helveticus*, (**D**) *E. faecalis*, (**E**) *S. thermophilus*, (**F**) *E. coli* Nissle in broth supplemented with 0.1%, 0.2%, 0.3%, or 0.4% CO. Bars marked with asterisks (*) indicate significant differences from the control (*p* < 0.05).

**Figure 2 animals-14-00023-f002:**
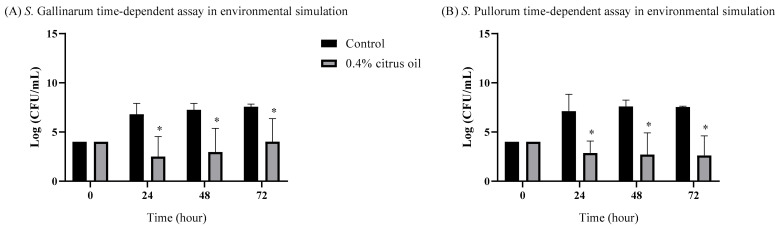
Growth performance of (**A**) *S. Gallinarum*, (**B**) *S. Pullorum*, in wooden chip as poultry floor simulation supplemented with 0.4% CO. Bars containing asterisks (*) are significantly different from the control (*p* < 0.05).

**Figure 3 animals-14-00023-f003:**
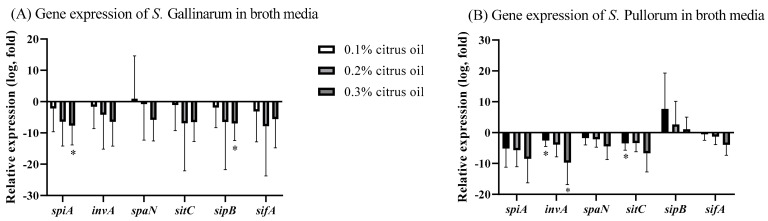
Gene expression of (**A**) *S. Gallinarum*, (**B**) *S. Pullorum* virulence genes after 24 h of treatment with either a control of LB broth supplemented with molecular water, or with 0.1%, 0.2%, 0.3% CO in LB broth. Asterisks (*) denote statistically significant differences from baseline.

**Figure 4 animals-14-00023-f004:**
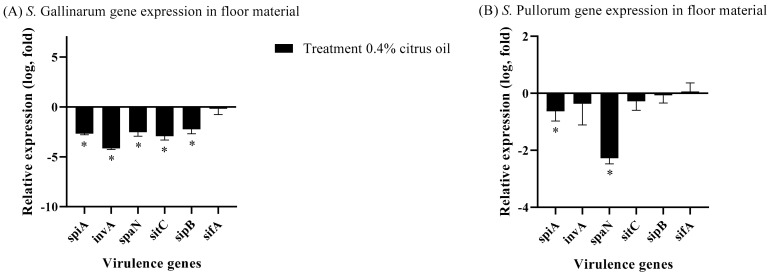
Gene expression of (**A**) *S. Gallinarum*, (**B**) *S. Pullorum* virulence genes after 24 h in wooden chip with either a control of water or treatment with 0.4% CO added. Asterisks (*) denote statistically significant differences from baseline.

**Table 1 animals-14-00023-t001:** Primers used in this study of virulence genes.

Primer	Primer Sequences (5′-3′)	Product Sizes (bp)	References
***spiA*-F**	CCAGGGGTCGTTAGTGTATTGCGTGAGATG	550	[33,34]
***spiA*-R**	CGCGTAACAAAGAACCCGTAGTGATGGATT		
***invA*-F**	CTGGCGGTGGGTTTTGTTGTCTTCTCTATT	1070	[33,34]
***invA*-R**	AGTTTCTCCCCCTCTTCATGCGTTACC		
***spaN*-F**	AAAAGCCGTGGAATCCGTTAGTGAAGT	504	[33,34]
***spaN*-R**	CAGCGCTGGGGATTACCGTTTTG		
***sitC*-F**	CAGTATATGCTCAACGCGATGTGGGTCTCC	768	[33,34]
***sitC*-R**	CGGGGCGAAAATAAAGGCTGTGATGAAC		
***sifA*-F**	TTTGCCGAACGCGCCCCCACACG	449	[33,34]
***sifA*-R**	GTTGCCTTTTCTTGCGCTTTCCACCCATCT		
***sipB*-F**	GGACGCCGCCCGGGAAAAACTCTC	875	[33,34]
***sipB*-R**	ACACTCCCGTCGCCGCCTTCACAA		

**Table 2 animals-14-00023-t002:** MIC and MBC of CO against *S. Gallinarum* and *S. Pullorum*.

Isolates	MIC	MBC
*S. Gallinarum* (CAT 375)	0.2%	0.2%
*S. Pullorum* (ATCC 13036)	0.4%	0.4%
*S. Gallinarum* (isolate from organic retail chicken sample)	0.4%	0.4%
*S. Pullorum* (isolate from farm sample)	0.4%	0.4%

## Data Availability

Data are contained within the article.

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
