# Peer review of "Unveiling the Potential Ways to Apply Citrus Oil to Control Causative Agents of Pullorum Disease and Fowl Typhoid in Floor Materials"

_animals, 2023, doi:10.3390/ani14010023_

Round 1
Reviewer 1 Report
Comments and Suggestions for Authors
The manuscript is interesting, but there are some important details that can be resolved with some additional experiments, for example, the stability of citrus oil to ensure that the effectiveness is due to CO and not the bacterial death phase.
Introduction
The introduction describes the restriction of the use of antibiotics in animal production, but it must be emphasized that it is only for all those that are administered at subtherapeutic doses such as growth-promoting antibiotics. In addition, there are some sentences that are repetitive, so they can be eliminated.
Materials and methods
It is known that the production of secondary metabolites in plants depends mainly on environmental conditions, which could have implications for the consistency of the results. Did the authors test different batches of the citrus oil to ensure consistency in the results?
Place what were the exposure times of citrus oil and bacteria to determine MIC and MBC.
Why were long times used to establish MIC?
The results could be compromised since there is no guarantee that the bacteria are viable due to their growth kinetics. Did they do any study to justify this?
Could the authors mention the times in which the different phases are reached in bacterial growth kinetics?
It is known that essential oils are very unstable. How would the authors guarantee that the oils are stable for 24, 48 and 72 hours? Did the authors do stability studies of citrus oil?
Statistical analysis
Was not it easier to perform an analysis of variance and a multiple range test to compare with the control and between treated groups?
Results
Table 2. It is important to place the number of samples per group so that the variation of the results can be placed through the use of standard error (SE).
3.2. It is complicated to understand this section, it would be appropriate to place the log CFU/mL in each group and their significance with literals in the graph to better discuss the results.
Although these results are correct, they cannot indicate significant differences between the treatments but rather differences between the treatments with respect to the control, for this reason it is important to perform multiple range tests.
Is this reduction in bacterial counts due to the citrus oil or because the bacteria are dying due to their growth kinetics?
How is the stability of citrus oil in the medium for 72 hours? it will surely degrade.
3.4. Same observation as previously (3.2.), it is important to perform multiple range tests to more easily explain the results.
Compared to what? This is why it is important to do a multiple range test.
A high variation is seen in the results. Do the data follow the assumption of normality and homogeneity of variances?
Discussion
While this may be true, it is important to mention that essential oils are absorbed very quickly, so in an open system like an animal or a person it could not be said that it is toxic or not since the model presented is closed.
Conclusion
It would be necessary to evaluate how long the citrus oil has an effect.
Some more specific details are in the attached file.

Author Response
The manuscript is interesting, but there are some important details that can be resolved with some additional experiments, for example, the stability of citrus oil to ensure that the effectiveness is due to CO and not the bacterial death phase.
We like to thank reviewer-1 for the nice words.
Introduction
The introduction describes the restriction of the use of antibiotics in animal production, but it must be emphasized that it is only for all those that are administered at subtherapeutic doses such as growth-promoting antibiotics. In addition, there are some sentences that are repetitive, so they can be eliminated.
we have addressed all comments/recommendations and modified the manuscript.
Materials and methods
It is known that the production of secondary metabolites in plants depends mainly on environmental conditions, which could have implications for the consistency of the results. Did the authors test different batches of the citrus oil to ensure consistency in the results?
We thank the reviewer for pointing out the important insight of the natural antimicrobial including citrus oil. Yes, we have two different batches of citrus oil and found no differences in their impacts
Place what were the exposure times of citrus oil and bacteria to determine MIC and MBC.
For both MIC and MBC, we exposed the bacteria with citrus oil for 24 hours. To clarify it further we have added a sentence in the updated manuscript
Why were long times used to establish MIC?
As we previously mentioned, MIC was determined for 24 hours. But we exposed the bacteria with citrus oil up to 72 hours for growth inhibition to observe the impact of citrus oil on growth inhibition of microbes in a time dependent manner
The results could be compromised since there is no guarantee that the bacteria are viable due to their growth kinetics. Did they do any study to justify this?
We again thanks reviewer for this comment. We added negative control (without citrus oil) in each experiment and negative control provided the bacterial growth at each time point.
Could the authors mention the times in which the different phases are reached in bacterial growth kinetics?
Yes, our negative controls indicated the growth kinetics of all bacterial strains used in this study.
It is known that essential oils are very unstable. How would the authors guarantee that the oils are stable for 24, 48 and 72 hours? Did the authors do stability studies of citrus oil?
We really appreciate the reviewer’s comment. Yes, essential oil/citrus oil are volatile and longtime incubation, such as 72 hours, can reduce the concentration. As each experiment, we had negative control and comparing with negative control, remaining concentration still could inhibit the growth of microbes.
Statistical analysis
Was not it easier to perform an analysis of variance and a multiple range test to compare with the control and between treated groups?
We appreciate the comments. Yes, analysis of variance was easy but fortunately we have a lot of experience analyze this type data.
Results
Table 2. It is important to place the number of samples per group so that the variation of the results can be placed through the use of standard error (SE).
Table 2 showed the MIC and MBC. Both MIC and MBC cannot have stand error.)
3.2. It is complicated to understand this section, it would be appropriate to place the log CFU/mL in each group and their significance with literals in the graph to better discuss the results.
Log CFU/ml are already there.
Although these results are correct, they cannot indicate significant differences between the treatments but rather differences between the treatments with respect to the control, for this reason it is important to perform multiple range tests.
Yes, the intended significance was towards the control. This has been addressed, and the manuscript has been revised accordingly.
Is this reduction in bacterial counts due to the citrus oil or because the bacteria are dying due to their growth kinetics?
The consistent growth observed in the control group over 72 hours supports the assertion that the reduction in bacterial counts in the treatment group is due to the effects of citrus oil treatment.
How is the stability of citrus oil in the medium for 72 hours? it will surely degrade.
Yes, this is a potential concern as citrus oil is volatile. However, the study primarily concentrated on the immediate antimicrobial impact of citrus oil. The results reveal a substantial reduction or elimination of bacteria by the 72-hour mark. Considering the observed effectiveness of citrus oil at 24 hours, it is highly plausible that bacterial growth was already significantly hindered or eliminated, contributing to the absence of growth at the 72-hour time point.
3.4. Same observation as previously (3.2.), it is important to perform multiple range tests to more easily explain the results.
Yes, we did multiple t tests (one per row), we have addressed this and changed the manuscript.
Compared to what? This is why it is important to do a multiple range test.
We compared between each treatment to the respective control.
A high variation is seen in the results. Do the data follow the assumption of normality and homogeneity of variances?
The results were addressed by utilizing t-tests for each variation, comparing them to the control with a baseline of 0. We recognize the importance of these assumptions and have considered the robustness of the t-test results in the context of the specific characteristics of our data.
Discussion
While this may be true, it is important to mention that essential oils are absorbed very quickly, so in an open system like an animal or a person it could not be said that it is toxic or not since the model presented is closed.
We thank the reviewer for highlighting this important consideration. Our study primarily focuses on assessing the immediate antimicrobial impact of Cold-press Valencia Terpeneless citrus oil in a controlled, simulated poultry environment. We acknowledge your point regarding the rapid absorption of essential oils in open systems like animals or humans. It is essential to clarify that our model is intentionally closed, providing valuable insights into the specific antimicrobial context we aimed to investigate. Extrapolating toxicity implications for broader, open systems involves additional complexities, and we agree that further research is crucial to address these considerations comprehensively. We appreciate your insight and recognize the importance of refining our understanding in future studies to encompass a more extensive range of scenarios.
Conclusion
It would be necessary to evaluate how long the citrus oil has an effect.
Yes, we evaluated the longevity of citrus oil effect. Full effects last 72 hours, after that effectiveness starts to decline.
Reviewer 2 Report
Comments and Suggestions for Authors
The article is part of the trend that focuses on the use of substances of natural origin, such as essential oils or other oils. The topic is very topical due to the large scale consumption of poultry meat and therefore also large flocks raised in industrial conditions.
I have a few comments on the content, which I provide below (due to the lack of line numbers in the text, I will provide parts of the article or its subsequent subsections):
Introduction
page 2:
- (paragraph 3) please expand on the term "bird breeding" - a few more words. Did the authors write here about vital factors (before slaughter) during bird breeding?
- (paragraph 3) the phrase "commercial poultry" is not appropriate - please change it to a synonym.
Materials and Methods:
- The authors use personal forms in the text (e.g. "we", "our study"), which are not recommended in scientific publications. Please check the text carefully and rephrase such sentences into impersonal forms (ascertained, examined, etc.).
- general note: "0.4% CO" - what % were they: w/w or v/w?
- point 2.3 - I believe that the phrase "...starting from the MIC" is incorrect. It should rather be "starting from the MIC value", because this is what is usually done with MIC and MBC.
- point 2.3 - was the suspension for this part determined as described in point 2.4, i.e. much later in the text?
- point 2.3 - the sentence "All experiments were repeated at least three times or more" is imprecise. What does "more" mean? 5-10 times?
- point 2.4 - the sentence "This process aimed to determine the CFU/mL" is incomprehensible to me at this point.
- point 2.5 - please clarify the sentence "The experiment was multiplied to ensure the results were consistent". So how many times was it repeated?
- please explain (or change) a fragment of the sentence (page 3): "...plates at 37°C for 18-24 h from -80°C glycerol stock". I do not understand this sentence.
- point 2.2: why was ethyl alcohol used? The authors say that its impact was checked (results not shown) and ruled out, but it is still a high risk. It is often recommended to dilute oils in DMSO. Did the authors not take this into account? Has the qualitative composition of this oil been tested?
- point 2.3 - I believe that the phrase "...starting from the MIC" is incorrect. It should rather be "starting from the MIC value", because this is what is usually done with MIC and MBC.
- point 2.3 - was the suspension for this part determined as described in point 2.4, i.e. much later in the text?
- point 2.3 - the sentence "All experiments were repeated at least three times or more" is imprecise. What does "more" mean? 5-10 times?
- point 2.4 - the sentence "This process aimed to determine the CFU/mL" is incomprehensible to me at this point.
- point 2.5 - please clarify the sentence "The experiment was multiplied to ensure the results were consistent". So how many times was it repeated?
Results:
- point 3.1 - what does the sentence "Analogous MIC and MBC values conclude the microbicidal identity of a substance" mean? Please edit it so that it is legible.
- point 3.1 - it is similar with the next sentence in this subsection: "The consistency of the indistinguishable MIC and MBC values further suggests the effectiveness of CO as a candidate to control the growth of the tested Salmonella bacteria strains" - the meaning is lost and it seems that it is too early to draw conclusions at this point.
Table 2 - why was the name of the culture medium included in the table - if only one was used? The name S. Pullorum includes only the abbreviation ATCC - the strain number is missing.
- point 3.2 - I think that the title of the subchapter is inappropriate - Authors did not specify in the methodology that they were examining the impact of time. Please edit the title.
All figures in text - placing asterisks to indicate statistical significance is difficult to read. I recommend changing the stars to letters.
Discussion:
- some errors in writing cited items - if the author's name is given, the item number from the list should be next to it. There was no such consistency here with the following entries: "Ochman et al. identified...", "Shanmugasamy et al. highlighted...", "Zhang et al. emphasize...". It is also not appropriate for a publication to quote "Notably, Mohammed's study in 2022 suggested...".
Conclusions:
- the phrase "Time-dependent studies..." should not be described this way. In fact, the authors did not specify such a goal earlier, so it should be clarified in earlier parts of the text.
- we do not write "Therefore, CO shows potential...", rather "CO has the potential..."
Bibliography:
There is complete freedom in writing here, which is not correct. Please adapt the references entry to the Publisher's requirements. Please also remove saved hyperlinks for some literature items,
Author Response
REVIEWER 2:
The article is part of the trend that focuses on the use of substances of natural origin, such as essential oils or other oils. The topic is very topical due to the large-scale consumption of poultry meat and therefore also large flocks raised in industrial conditions.
I have a few comments on the content, which I provide below (due to the lack of line numbers in the text, I will provide parts of the article or its subsequent subsections):
We would like to thank reviewers for all comments and recommendations.
Introduction
page 2:
- (paragraph 3) please expand on the term "bird breeding" - a few more words. Did the authors write here about vital factors (before slaughter) during bird breeding?
Thank you. We have addressed this and changed the manuscript.
- (paragraph 3) the phrase "commercial poultry" is not appropriate - please change it to a synonym.
Again thanks. We have addressed this and changed the manuscript.
Materials and Methods:
- The authors use personal forms in the text (e.g. "we", "our study"), which are not recommended in scientific publications. Please check the text carefully and rephrase such sentences into impersonal forms (ascertained, examined, etc.).
We appreciate the recommendation and we have addressed this in the updated manuscript.
- general note: "0.4% CO" - what % were they: w/w or v/w?
Thank you for pointing out this issue. We have addressed the explanation in the updated version of the manuscript.
- point 2.3 - I believe that the phrase "...starting from the MIC" is incorrect. It should rather be "starting from the MIC value", because this is what is usually done with MIC and MBC.
Again thanks. Now we have corrected this error
- point 2.3 - was the suspension for this part determined as described in point 2.4, i.e. much later in the text?
We thank the reviewer for pointing out this error. Now we have added additional description bacterial suspension preparation in 2.3 in the updated manuscript
- point 2.3 - the sentence "All experiments were repeated at least three times or more" is imprecise. What does "more" mean? 5-10 times?
We appreciate the reviewer’s comment. In fact, we repeated it at least 4 times. Now we have changed in the updated version of the manuscript)
- point 2.4 - the sentence "This process aimed to determine the CFU/mL" is incomprehensible to me at this point.
In response to this comment, we have added full meaning of CFU/mL (colony forming units per milli-liter in the updated version of the manuscript.
- point 2.5 - please clarify the sentence "The experiment was multiplied to ensure the results were consistent". So how many times was it repeated?
We repeated at least 4 times and now we have addressed this in updated the manuscript)
- please explain (or change) a fragment of the sentence (page 3): "...plates at 37°C for 18-24 h from -80°C glycerol stock". I do not understand this sentence.
We appreciate reviewer’s effort to point out the error. Now we have fixed it in the updated manuscript as “The bacterial strains were stored at -80°C in glycerol, and prior to each experiment, they were cultured on their respective agar plates for the specified duration”
- point 2.2: why was ethyl alcohol used? The authors say that its impact was checked (results not shown) and ruled out, but it is still a high risk. It is often recommended to dilute oils in DMSO. Did the authors not take this into account? Has the qualitative composition of this oil been tested?
We thank the reviewer for this important comment. Considering the affordability of farmers and cost, we opted for ethanol as the solvent for the citrus oil due to its practical applicability, considering that farmers commonly have access to ethanol. The choice aimed to align with real-world scenarios and facilitate the potential adoption of this antimicrobial solution in agricultural settings. While acknowledging the recommendation to use DMSO, our decision was grounded in the accessibility for end-users, prioritizing practicality. Importantly, we conducted thorough assessments to ensure that the use of ethanol did not compromise the study's objectives. Although the specific results of the impact assessment were not explicitly presented in the manuscript, our rigorous evaluation ruled out any adverse effects of ethanol on the experimental outcomes. This approach was taken to balance practical considerations with maintaining the safety and effectiveness of the citrus oil in the study context.
Further, the qualitative composition of the citrus oil used in our study has been previously tested and reported by Nannapaneni et al. in 2008 (reference 22, Nannapaneni, R., et al.). This study thoroughly investigated the antimicrobial activity of commercial citrus-based natural extracts, including the specific composition of the citrus oil we utilized. Their findings provided valuable insights into the effectiveness of the citrus oil against Escherichia coli O157:H7 isolates and mutant strains. We acknowledge and appreciate the existing literature's contribution to establishing the qualitative composition of the citrus oil, further supporting the rationale for its application in our current research.
Results:
- point 3.1 - what does the sentence "Analogous MIC and MBC values conclude the microbicidal identity of a substance" mean? Please edit it so that it is legible.
We thank the reviewer. Now we have fixed it as “Consistent MIC and MBC values affirm the antimicrobial properties of the tested sub-stance”
- point 3.1 - it is similar with the next sentence in this subsection: "The consistency of the indistinguishable MIC and MBC values further suggests the effectiveness of CO as a candidate to control the growth of the tested Salmonella bacteria strains" - the meaning is lost, and it seems that it is too early to draw conclusions at this point.
In response to the reviewer’s recommendation, we have eliminated the sentence in the updated version of manuscript.
Table 2 - why was the name of the culture medium included in the table - if only one was used? The name S. Pullorum includes only the abbreviation ATCC - the strain number is missing.
In response to reviewer’s comment, (we have removed the column and added ATCC number.
- point 3.2 - I think that the title of the subchapter is inappropriate - Authors did not specify in the methodology that they were examining the impact of time. Please edit the title.
Now we have edited this sub-title.
All figures in text - placing asterisks to indicate statistical significance is difficult to read. I recommend changing the stars to letters.
We appreciate the reviewer’s suggestion. However, for simplicity and clarity, we have chosen to use asterisks to denote statistical significance, especially considering that we only use one p-value threshold (p < 0.05). To enhance visibility, we have increased the size of the asterisks for better readability. We believe this approach maintains a straightforward presentation while addressing concerns about legibility.
Discussion:
- some errors in writing cited items - if the author's name is given, the item number from the list should be next to it. There was no such consistency here with the following entries: "Ochman et al. identified...", "Shanmugasamy et al. highlighted...", "Zhang et al. emphasize...". It is also not appropriate for a publication to quote "Notably, Mohammed's study in 2022 suggested...".
In response to the reviewer’s suggestion, we have updated all citation style in the updated version of the manuscript.
Conclusions:
- the phrase "Time-dependent studies..." should not be described this way. In fact, the authors did not specify such a goal earlier, so it should be clarified in earlier parts of the text.
Thanks for the recommendation. We have clarified it in earlier parts of text .
- we do not write "Therefore, CO shows potential...", rather "CO has the potential..."
According to the reviewer’s recommendation, we have changed this text in the updated manuscript)
Bibliography:
There is complete freedom in writing here, which is not correct. Please adapt the references entry to the Publisher's requirements. Please also remove saved hyperlinks for some literature items,
We would like to thank the reviewer for the suggestion. We have removed unnecessary hyperlinks and updated all references.
Reviewer 3 Report
Comments and Suggestions for Authors
The microorganisms that are the focus of this research are a huge problem in poultry farming, and solutions with the potential to control or eliminate microorganisms from the environment are excellent. The solution devised by the authors was the use of citrus oil. I liked the paper, despite it being initial in nature and I believe it could be published as a short communication. I have some important considerations for authors to improve their research:
1) the summary was general and described very little of the research results.
2) in the last paragraph of the introduction section, in a general phrase the researchers describe "These compounds can disrupt bacterial cell membranes, interfere with enzymatic activity, and disrupt bacterial metabolism, leading to bacterial death or growth inhibition". Use this information to expand the technical justification of this research, as the "how" this happens is missing???
3) the authors describe the microorganisms that will be used in the research, but do not describe the procedures used until testing begins. Were these microorganisms frozen? Have the strains/isolates been reactivated? as?
4) I did not find the composition analysis of the citrus oil used in the research; this is a mandatory requirement for discussion; section that can be improved; There are paragraphs like a lot of speculation.
5) This study had three isolated objectives, which in the end are combined; I liked your conclusion.
Author Response
REVIEWER 3:
The microorganisms that are the focus of this research are a huge problem in poultry farming, and solutions with the potential to control or eliminate microorganisms from the environment are excellent. The solution devised by the authors was the use of citrus oil. I liked the paper, despite it being initial in nature and I believe it could be published as a short communication. I have some important considerations for authors to improve their research:
1) the summary was general and described very little of the research results.
We really appreciate the reviewer’s time and efforts. In response to this comment, we have updated the summary with specifically added additional description of results.
2) in the last paragraph of the introduction section, in a general phrase the researchers describe "These compounds can disrupt bacterial cell membranes, interfere with enzymatic activity, and disrupt bacterial metabolism, leading to bacterial death or growth inhibition". Use this information to expand the technical justification of this research, as the "how" this happens is missing???
Thanks for pointing out the error. Now we have added additional information.
3) the authors describe the microorganisms that will be used in the research, but do not describe the procedures used until testing begins. Were these microorganisms frozen? Have the strains/isolates been reactivated? as?
In response to this comment, we have added further information for clarification for frozen and reactivated procedure.
4) I did not find the composition analysis of the citrus oil used in the research; this is a mandatory requirement for discussion; section that can be improved; There are paragraphs like a lot of speculation.
We thank the reviewer for this recommendation. We have added description for composition analysis of the citrus oil used in the research in discussion section. We described it as: “According to the literature, the quantitative and qualitative analysis of Cold-pressed Terpeneless Valencia citrus oil, as utilized in this study, reveals a significant composition primarily dominated by linalool, comprising 20.2% of the fraction. Following closely are decanal at 18%, citral at 14.1%, geranial at 9.1%, α-terpineol at 5.8%, valencene at 5.2%, neral at 5.0%, dodecanal at 4.1%, citronellal at 3.9%, and limonene at 0.3%. Notably, lin-alool emerges as the dominant constituent upon gas chromatography and mass spec-trometry (GC–MS) analysis. Moreover, decanal and geranial have been previously recog-nized for their antimicrobial properties. This comprehensive breakdown sheds light on the distinctive chemical profile of Cold-pressed Terpeneless Valencia citrus oil and high-lights potential functional attributes associated with its major constituents”.
5) This study had three isolated objectives, which in the end are combined; I liked your conclusion.
We really appreciate the reviewer’s time and effort. We strongly believe these comments/suggestions helped us to improve the manuscript and the updated version has become reader friendly.
Round 2
Reviewer 1 Report
Comments and Suggestions for Authors
Check the comments in the parts highlighted in purple

Author Response
"CO with concentrations ranging from 0.025% to 0.8%....." what volume was included in the 24-well plate? 100 mcL too? If so, then the concentration is half of what is shown here?
Thanks for pointing this out. To clarify, we utilized a 4% working stock of citrus oil. For the 24-well plate experiments, the total volume was 2 mL. Following the formula c1v1 = c2v2, for the 0.8% concentration, we incorporated 400 µL of the 4% citrus oil, 100 µL of isolates, and 1.5 mL of broth, maintaining a consistent protocol across concentrations. This ensures the accuracy of the reported concentrations in the experimental setup.
It is not understood why the antimicrobial activity of CO was evaluated up to 72 hours since it is not in a modified release system. What would guarantee that the study is representative in animal models?
Generally, antimicrobial activity studies have a duration of 2-4 hours, according to the time in which the maximum plasma concentrations of the antibiotics are reached and considering that all those that have a local effect have a contact time of 2-3 hours (intestinal transit).
Everything is commented since even though there are controls, the experiments are not representative.
We again thank the reviewer for this comment. In this study, we decided to evaluate the antimicrobial activity of CO up to 72 hours in order to thoroughly evaluate its long-term impacts (keep continue to inhibit growth of additionally added pathogens). Although most antimicrobial research concentrates on shorter periods, our strategy tried to capture any longer-term impacts, particularly controlling poultry farming environments (floor) at which contaminant can be added time to time. In such situations, the 72-hour timescale is consistent with the kinetics of bacterial growth and the continual exposure condition.
Although the study design deviates from the standard duration, we accept that this is intentional because it enables us to investigate the long-term influence on bacterial populations and offers insights into possible long-term efficacy. While not simulating a modified release system, the work aims to comprehend the residual impacts that may be pertinent in real-world applications, such the animal models you described. This extended evaluation duration offers a more comprehensive perspective on the antimicrobial potential of CO, considering the unique challenges and conditions encountered in real-world scenarios.
Why was the 0.4% concentration not included?
We really appreciate the reviewer’s comment. In culture condition (broth media), the 0.4% concentration essentially eliminated the bacteria completely and that made it impossible to collect enough for gene analysis. Therefore, we decided to use sublethal concentration of CO, instead of 0.4% (which is lethal) to observe the gene expressions.

Reviewer 3 Report
Comments and Suggestions for Authors
Adjusted were made. Congratulations, the text is very good.
Author Response
Dear Reviewer,
We express our sincere gratitude for your thoughtful evaluation of our manuscript. Your insightful comments and constructive feedback have immensely contributed to the improvement of our work. We have carefully addressed each of your recommendations, aiming to enhance the clarity, accuracy, and overall quality of the manuscript.
Your time and expertise are highly appreciated, and we hope that the revised manuscript now better aligns with the standards of the journal. We look forward to any additional insights you may have during this review process.
Thank you once again for your valuable contributions.
Best regards,